# SafeDiffuser: Safe Planning with Diffusion Probabilistic Models

## Abstract

Diffusion model-based approaches have shown promise in data-driven planning. Although these planners are typically used in decision-critical applications, there are yet no known safety guarantees established for them. In this paper, we address this limitation by introducing SafeDiffuser, a method to equip probabilistic diffusion models with safety guarantees via control barrier functions. The key idea of our approach is to embed finite-time diffusion invariance, i.e., a form of specification mainly consisting of safety constraints, into the denoising diffusion procedure. This way we enable data generation under safety constraints. We show that SafeDiffusers maintain the generative performance of diffusion models while also providing robustness in safe data generation. We finally test our method on a series of planning tasks, including maze path generation, legged robot locomotion, and 3D space manipulation, and demonstrate the advantages of robustness over vanilla diffusion models[1].

## 1 Introduction

Diffusion models (Sohl-Dickstein et al., 2015) (Ho et al., 2020) is a family of generative modeling approaches that enabled major breakthroughs in image synthesis (Dhariwal & Nichol, 2021) (Du et al., 2020b) (Saharia et al., 2022). Recently, diffusion models, termed diffusers (Janner et al., 2022), have shown promise in trajectory planning for a variety of robotic

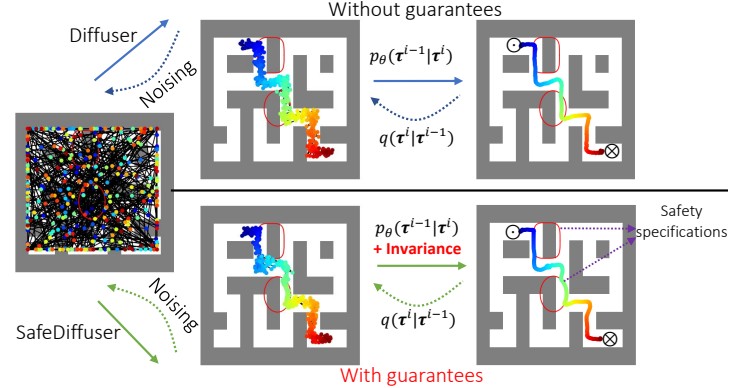

Figure 1: Our proposed SafeDiffuser (lower) generates safe trajectories with guarantees, while the diffuser (upper) fails (from ⊙ to ⊗).

tasks. Compared to existing planning methods, diffusers (a) enable long-horizon planning with multi-modal action distributions and stable training, (b) easily scale to high-dimensional trajectory planning, and (c) are flexible for behavior synthesis.

During inference, the diffuser, conditioned on the current state and objectives, starts from Gaussian noise to generate clean planning trajectories based on which we can get a control policy. After applying this control policy one step forward, we get a new state and rerun the diffusion procedure to get a new planning trajectory. This process is repeated until the objective is achieved.

Although the application of these planners is mainly in safety-critical applications, there are no known safety guarantees established for them. For instance, the planning trajectory could easily violate safety constraints in the maze (as shown in Fig. 1). This shortcoming demands a fundamental fix to diffusion models to ensure the safe generation of planning trajectories in safety-critical applications such as trustworthy policy learning (Xiao et al., 2023a).

---

[1]Videos can be viewed here: `https://safediffuser.github.io/safediffuser/`

In this paper, we propose to ensure diffusion models with specification guarantees using finite-time diffusion invariance. An invariance set is a form of specification mainly consisting of safety constraints in planning tasks. We ensure that diffusion models are invariant to uncertainties in terms of safety in the diffusion procedure. We achieve safety by combining receding horizon control with diffusion models. In receding horizon control, we compute safe paths incrementally. The key insight is to replace traditional path planning with diffusion-based path generation, allowing a broader exploration of the path space and making it relatively easy to include additional constraints. The computed path is combined with simulation to validate that it can be safely actuated.

To ensure diffusers with specifications guarantees, we first find diffusion dynamics for the denoising diffusion procedure. Then, we use control barrier functions (CBFs) (Ames et al., 2017) (Glotfelter et al., 2017) (Nguyen & Sreenath, 2016) (Xiao & Belta, 2019), to formally guarantee the satisfaction of specifications at the end of the diffusion procedure. CBFs work well in planning time using robot dynamics. However, doing this in diffusion models poses extra challenges since the generated data is not directly associated with robot dynamics which makes the use of CBFs non-trivial. As opposed to existing literature, (i) we propose to embed invariance into the diffusion time for diffusers. Thus, finite-time invariance is required in diffusers since specifications are usually violated as the trajectory is initially Gaussian noise. (ii) We propose to add diffusion time components in invariance to address local trap problems (i.e., trajectory points getting stuck at obstacle boundaries) that are prominent in planning. (iii) We propose a quadratic program approach to incorporate finite-time diffusion invariance into the diffusion to maximally preserve the performance.

In summary, we make the following **contributions**:

- We propose formal guarantees for diffusion probabilistic models via control-theoretic invariance.
- We propose a novel notion of finite-time diffusion invariance, and use a class of CBFs to incorporate it into the diffusion time of the procedure. We proposed three different safe diffusers, and show how we may address the local trap problem from specifications that are prominent in planning tasks.
- We demonstrate the effectiveness of our method on a variety of planning tasks using diffusion models, including safe planning in maze, robot locomotion, and manipulation.

## 2 PRELIMINARIES

In this section, we provide background on diffusion models and forward invariance in control theory.

**Diffusion Probabilistic Models.** Diffusion probabilistic models (Sohl-Dickstein et al., 2015; Ho et al., 2020; Janner et al., 2022) are a type of latent variable models. They describe the process of data generation as a series of iterative denoising steps. Here, the model is represented as $p_\theta(\tau^{i-1}|\tau^i)$, $i \in \{1, \ldots, N\}$, where $\tau^1, \ldots, \tau^N$ are latent variables mirroring the dimension of the original, noise-free data $\tau^0 \sim q(\tau^0)$, and $N$ signifies the total number of denoising steps. This denoising sequence is essentially the inverse of a forward diffusion process denoted as $q(\tau^i|\tau^{i-1})$ where the initial clean data is progressively degraded by adding noise. The process of generating data through denoising is expressed as (Janner et al., 2022):

$$p_\theta(\tau^0) = \int p_\theta(\tau^{0:N}) d\tau^{1:N} = \int p(\tau^N) \prod_{i=1}^{N} p_\theta(\tau^{i-1}|\tau^i) d\tau^{1:N}. \tag{1}$$

In this equation, $p(\tau^N)$ represents a standard Gaussian prior distribution. The joint distribution $p_\theta(\tau^{0:N})$ is defined as a Markov chain with learned Gaussian transitions that commence at $p(\tau^N)$ (Janner et al., 2022). The optimization parameter $\theta$ is achieved by minimizing the common variational bound on the negative log-likelihood of the reverse process, formalized as (Janner et al., 2022): $\theta^* = \arg\min_\theta \mathbb{E}_{\tau^0} \left[ -\log p_\theta(\tau^0) \right]$. The forward diffusion process, denoted as $q(\tau^i|\tau^{i-1})$, is typically predefined. Conversely, the reverse process is frequently characterized as a Gaussian process, featuring a mean and variance that vary depending on time.

**Notations.** For the sake of consistency, we keep our notations as that proposed in Janner et al. (2022) as follows: Here, two distinct 'times' are discussed: one associated with the diffusion process and the other with the planning horizon. These are differentiated as follows: superscripts (employing $i$

when unspecified) indicate the diffusion time of a trajectory or state, whereas subscripts (using $k$ when unspecified) denote the planning time of a state within the trajectory. For instance, $\boldsymbol{\tau}^0$ refers to the trajectory at the initial denoising diffusion time step, which is a noiseless trajectory. In a similar vein, $\boldsymbol{x}_k^0$ represents the state at the $k^{th}$ planning time step during the first denoising diffusion step, indicating a noiseless state. When clarity permits, we simplify this notation to $\boldsymbol{x}_k = \boldsymbol{x}_k^0$ (and similarly $\boldsymbol{\tau} = \boldsymbol{\tau}^0$). Moreover, a trajectory $\boldsymbol{\tau}^i$ is conceptualized as a sequence of states across planning time, articulated as $\boldsymbol{\tau}^i = (\boldsymbol{x}_0^i, \boldsymbol{x}_1^i, \ldots, \boldsymbol{x}_k^i, \ldots, \boldsymbol{x}_H^i)$, where $H \in \mathbb{N}$ defines the planning horizon.

**Forward Invariance in Control Theory.** Consider an affine control system of the form:

$$\dot{\boldsymbol{x}}_t = f(\boldsymbol{x}_t) + g(\boldsymbol{x}_t)\boldsymbol{u}_t \tag{2}$$

where $\boldsymbol{x}_t \in \mathbb{R}^n$, $f : \mathbb{R}^n \to \mathbb{R}^n$ and $g : \mathbb{R}^n \to \mathbb{R}^{n \times q}$ are locally Lipschitz, and $\boldsymbol{u}_t \in U \subset \mathbb{R}^q$, where $U$ denotes a control constraint set. $\dot{\boldsymbol{x}}_t$ denotes the (planning) time derivative of state $\boldsymbol{x}_t$.

**Definition 2.1.** *(Set invariance): A set $C \subset \mathbb{R}^n$ is forward invariant for system (2) if its solutions for some $\boldsymbol{u} \in U$ starting at any $\boldsymbol{x}_0 \in C$ satisfy $\boldsymbol{x}_t \in C, \forall t \geq 0$.*

**Definition 2.2.** *(Extended class $\mathscr{K}$ function): A Lipschitz continuous function $\alpha : [-b, a) \to (-\infty, \infty), b > 0, a > 0$ belongs to extended class $\mathscr{K}$ if it is strictly increasing and $\alpha(0) = 0$.*

Consider a safety constraint $b(\boldsymbol{x}_t) \geq 0$ for system (2), where $b : \mathbb{R}^n \to \mathbb{R}$ is continuously differentiable, we define a safe set in the form: $C := \{\boldsymbol{x}_t \in \mathbb{R}^n : b(\boldsymbol{x}_t) \geq 0\}$.

**Definition 2.3.** *(Control Barrier Function (CBF) (Ames et al., 2017)): A function $b : \mathbb{R}^n \to \mathbb{R}$ is a CBF if there exists an extended class $\mathscr{K}$ function $\alpha$ such that*

$$\sup_{\boldsymbol{u}_t \in U} \left[ L_f b(\boldsymbol{x}_t) + [L_g b(\boldsymbol{x}_t)]\boldsymbol{u}_t + \alpha(b(\boldsymbol{x}_t)) \right] \geq 0, \tag{3}$$

*for all $\boldsymbol{x}_t \in C$. $L_f$ and $L_g$ denote Lie derivatives w.r.t. $\boldsymbol{x}$ along $f$ and $g$, respectively.*

**Theorem 2.4** ((Ames et al., 2017)). *Given a CBF $b(\boldsymbol{x}_t)$ from Def. 2.3, if $\boldsymbol{x}_0 \in C$, then any Lipschitz continuous controller $\boldsymbol{u}_t$ that satisfies (3), $\forall t \geq 0$ renders $C$ forward invariant for system (2).*

## 3 SAFE DIFFUSER

In this section, we propose three different safe diffusers to ensure the safe generation of data in diffusion, i.e., to ensure the satisfaction of specifications $b(\boldsymbol{x}_k) \geq 0, \forall k \in \{0, \ldots, H\}$. Each of the proposed safe diffusers has its own flexibility, such as avoiding local traps in planning. We consider discretized system states in the sequel. Safety in continuous planning time can be guaranteed using a lower hierarchical control framework employing other CBFs, as in (Ames et al., 2017; Nguyen & Sreenath, 2016; Xiao & Belta, 2019).

In the denoising diffusion procedure, since the learned Gaussian transitions start at $p(\boldsymbol{x}^N) \sim \mathcal{N}(0, \boldsymbol{I})$, it is highly likely that specifications are initially violated, i.e., $\exists k \in \{0, \ldots, H\}, b(\boldsymbol{x}_k^N) < 0$. For safe data generation, we wish to have $b(\boldsymbol{x}_k^0) \geq 0 (i.e., b(\boldsymbol{x}_k) \geq 0), \forall k \in \{0, \ldots, H\}$. Since the maximum denoising diffusion step $N$ is limited, this needs to be guaranteed in a finite diffusion time step. Therefore, we propose the finite-time diffusion invariance of the diffusion procedure as follows:

**Definition 3.1** (Finite-time Diffusion Invariance). *If there exists $i \in \{0, \ldots, N\}$ such that $b(\boldsymbol{x}_k^j) \geq 0, \forall k \in \{0, \ldots, H\}, \forall j \leq i$, then a denoising diffusion procedure $p_\theta(\boldsymbol{\tau}^{i-1}|\boldsymbol{\tau}^i), i \in \{1, \ldots, N\}$ with respect to a specification $b(\boldsymbol{x}_k) \geq 0, \forall k \in \{0, \ldots, H\}$ is finite-time diffusion invariant.*

The above definition can be interpreted as that if $b(\boldsymbol{x}_k^N) \geq 0, k \in \{0, \ldots, H\}$, then we require $b(\boldsymbol{x}_k^i) \geq 0, \forall i \in \{0, \ldots, N\}$ (similar to the forward invariance definition as in Def. 2.1); otherwise, we require that $b(\boldsymbol{x}_k^j) \geq 0, \forall j \in \{0, \ldots, i\}, i \in \{0, \ldots, N\}$, where $i$ is a finite diffusion time.

In the following, we propose three methods for finite-time diffusion invariance. The first method is a general form of the safe-diffuser, and the other two are variants to address local traps in planning.

### 3.1 ROBUST-SAFE DIFFUSER

The safe denoising diffusion procedure is considered at every diffusion step. Following (1), the data generation at the diffusion time $j \in \{0, \ldots, N-1\}$ is given by:

$$p_\theta(\boldsymbol{\tau}^j) = \int p(\boldsymbol{\tau}^N) \prod_{i=j+1}^{N} p_\theta(\boldsymbol{\tau}^{i-1}|\boldsymbol{\tau}^i) d\boldsymbol{\tau}^{j+1:N} \tag{4}$$

A sample $\boldsymbol{\tau}^j, j \in \{0, \ldots, N-1\}$ follows the data distribution in (4), i.e., we have

$$\boldsymbol{\tau}^j \sim p_\theta(\boldsymbol{\tau}^j). \tag{5}$$

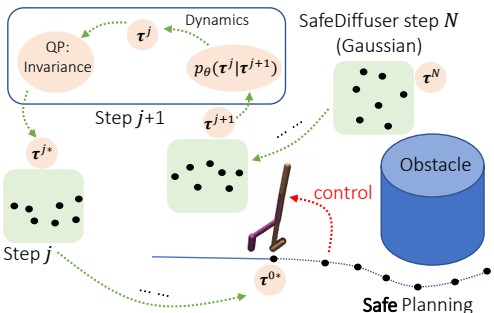

Figure 2: SafeDiffuser workflow. SafeDiffuser performs an additional step of invariance QP solver.

The denoising diffusion dynamics are then given by:

$$\dot{\boldsymbol{\tau}}^j = \lim_{\Delta\tau \to 0} \frac{\boldsymbol{\tau}^j - \boldsymbol{\tau}^{j+1}}{\Delta\tau} \tag{6}$$

where $\dot{\boldsymbol{\tau}}$ is the (diffusion) time derivative of $\boldsymbol{\tau}$. $\Delta\tau > 0$ is a small enough diffusion time step length during implementations, and $\boldsymbol{\tau}^{j+1}$ is available from the last diffusion step.

In order to impose finite-time diffusion invariance on the diffusion procedure, we wish to make diffusion dynamics (6) controllable. We reformulate (6) as

$$\dot{\boldsymbol{\tau}}^j = \boldsymbol{u}^j, \tag{7}$$

where $\boldsymbol{u}^j$ is a control variable of the same dimensionality as $\boldsymbol{\tau}^j$. On the other hand, we wish $\boldsymbol{u}^j$ to stay close to $\frac{\boldsymbol{\tau}^j - \boldsymbol{\tau}^{j+1}}{\Delta\tau}$ in order to maximally preserve the performance of the diffusion model. The above model can be rewritten in terms of each state on the trajectory $\boldsymbol{\tau}^j$: $\dot{x}_k^j = u_k^j$, where $u_k^j$ is the $k^{th}$ component of $\boldsymbol{u}^j$. Then, we can define CBFs to ensure the satisfaction of $b(x_k^j) \geq 0$ (in finite diffusion time): $h(u_k^j|x_k^j) := \frac{db(x_k^j)}{dx_k^j} u_k^j + \alpha(b(x_k^j)) \geq 0, k \in \{0, \ldots, H\}, j \in \{0, \ldots, N-1\}$, where $\alpha(\cdot)$ is an extended class $\mathscr{K}$ function. We have the following theorem to show the finite-time diffusion invariance (proof is given in Appendix, recall that $H$ is the planning horizon, $N$ is the diffusion step):

**Theorem 3.2.** *Let the diffusion dynamics be defined as in (6) whose controllable form is defined as in (7). If the robust term $\gamma: \mathbb{R}^2 \to \mathbb{R}$ is chosen such that $\gamma(N, \varepsilon) \geq |\gamma(N, \varepsilon) - b(x_k^N)|e^{-\varepsilon N}, \forall k \in \{0, \ldots, H\}$ and*

$$h(u_k^j|x_k^j) \geq 0, \forall k \in \{0, \ldots, H\}, \forall j \in \{0, \ldots, N-1\}, \tag{8}$$

*where $h(u_k^j|x_k^j) = \frac{db(x_k^j)}{dx_k^j} u_k^j + \varepsilon(b(x_k^j) - \gamma(N, \varepsilon)), \varepsilon > 0$ correponds to a linear class $\mathscr{K}$ function in CBF (3), then the diffusion procedure $p_\theta(\boldsymbol{\tau}^{i-1}|\boldsymbol{\tau}^i), i \in \{1, \ldots, N\}$ is finite-time diffusion invariant.*

One possible issue in the robust-safe diffusion is that if $b(x_k^j) \geq 0$ when $j$ is close to the diffusion step $N$, then $x_k^j$ can never violate the specification after diffusion step $j$. The state $x_k^j$ may get stuck at local traps from specifications during diffusion (see Fig. 7 of appendix). In order to address this issue, we propose a relaxed-safe diffuser and a time-varying-safe diffuser in the following subsections.

## 3.2 RELAXED-SAFE DIFFUSER

In order to address the local trap problems imposed by specifications during the denoising diffusion procedure, we propose a variation of the robust-safe diffuser. We define the diffusion dynamics and their controllable form as in (6) - (7). The modified versions for CBFs are in the form:

$$h(u_k^j, r_k^j|x_k^j) := \frac{db(x_k^j)}{dx_k^j} u_k^j + \alpha(b(x_k^j)) - w_k(j)r_k^j \geq 0, k \in \{0, \ldots, H\}, j \in \{0, \ldots, N-1\}, \tag{9}$$

where $r_k^j \in \mathbb{R}$ is a relaxation variable that is to be determined (shown in the next section). $w_k(j) \geq 0$ is a diffusion time-varying weight on the relaxation variable such that it gradually decreases to 0 as $j \to N_0, 0 \leq N_0 \leq N-1$, and $w_k(j) = 0$ for all $j \leq N_0$.

When $w_k(j)$ decreases to 0, the condition (9) becomes a hard constraint. We also have the following theorem to show the finite-time diffusion invariance (proof is given in appendix):

**Theorem 3.3.** *Let the diffusion dynamics be defined as in (6) whose controllable form is defined as in (7). If the robust term $\gamma : \mathbb{R}^2 \to \mathbb{R}$ is chosen such that $\gamma(N_0, \varepsilon) \geq |\gamma(N_0, \varepsilon) - b(x_k^{N_0})|e^{-\varepsilon N_0}, \forall k \in \{0, \ldots, H\}, 0 \leq N_0 \leq N - 1$ and there exists a time-varying $w_k(j)$ where $w_k(j) = 0$ for all $j \leq N_0$ s.t.*

$$h(\boldsymbol{u}_k^j, r_k^j | \boldsymbol{x}_k^j) \geq 0, \forall k \in \{0, \ldots, H\}, \forall j \in \{0, \ldots, N-1\}, \tag{10}$$

*where $h(\boldsymbol{u}_k^j, r_k^j | \boldsymbol{x}_k^j) = \frac{db(\boldsymbol{x}_k^j)}{d\boldsymbol{x}_k^j} \boldsymbol{u}_k^j + \varepsilon(b(\boldsymbol{x}_k^j) - \gamma(N_0, \varepsilon)) - w_k(j) r_k^j, \varepsilon > 0$ corresponds to a linear class $\mathcal{K}$ function in CBF (3), then the diffusion procedure $p_\theta(\boldsymbol{\tau}^{i-1} | \boldsymbol{\tau}^i), i \in \{0, \ldots, N\}$ is finite-time diffusion invariant.*

### 3.3 TIME-VARYING-SAFE DIFFUSER

As an alternative to the relaxed-safe diffuser, we propose another safe diffuser called the time-varying-safe diffuser in this subsection. The proposed time-varying-safe diffuser can also address the local trap issues induced by specifications.

In this case, we directly modify the specification $b(\boldsymbol{x}_k^j) \geq 0$ by a diffusion time-varying function $\gamma_k : j \to \mathbb{R}$ (as opposed to the last two safe diffusers with a constant robust term $\gamma(N, \varepsilon)$) in the form:

$$b(\boldsymbol{x}_k^j) - \gamma_k(j) \geq 0, k \in \{0, \ldots, H\}, j \in \{0, \ldots, N\}, \tag{11}$$

where $\gamma_k(j)$ is continuously differentiable, and is defined such that $\gamma_k(N) \leq b(\boldsymbol{x}_k^N)$ and $\gamma_k(0) = 0$.

The modified time-varying specification can then be enforced using CBFs: $h(\boldsymbol{u}_k^j | \boldsymbol{x}_k^j, \gamma_k(j)) := \frac{db(\boldsymbol{x}_k^j)}{d\boldsymbol{x}_k^j} \boldsymbol{u}_k^j - \dot{\gamma}_k(j) + \alpha(b(\boldsymbol{x}_k^j) - \gamma_k(j)) \geq 0, k \in \{0, \ldots, H\}, j \in \{0, \ldots, N-1\}$, where $\dot{\gamma}_k(j)$ is the diffusion time derivative of $\gamma_k(j)$. Finally, we have the following theorem to show the finite-time diffusion invariance (proof is given in Appendix):

**Theorem 3.4.** *Let the diffusion dynamics be defined as in (6) whose controllable form is defined as in (7). If there exist an extended class $\mathcal{K}$ function $\alpha$ and a time-varying function $\gamma_k(j)$ where $\gamma_k(N) \leq b(\boldsymbol{x}_k^N)$ and $\gamma_k(0) = 0$ such that*

$$h(\boldsymbol{u}_k^j | \boldsymbol{x}_k^j, \gamma_k(j)) \geq 0, \forall k \in \{0, \ldots, H\}, \forall j \in \{0, \ldots, N-1\}, \tag{12}$$

*where $h(\boldsymbol{u}_k^j | \boldsymbol{x}_k^j, \gamma_k(j)) = \frac{db(\boldsymbol{x}_k^j)}{d\boldsymbol{x}_k^j} \boldsymbol{u}_k^j - \dot{\gamma}_k(j) + \alpha(b(\boldsymbol{x}_k^j) - \gamma_k(j))$, then the diffusion procedure $p_\theta(\boldsymbol{\tau}^{i-1} | \boldsymbol{\tau}^i), i \in \{0, \ldots, N\}$ is finite-time diffusion invariant.*

## 4 ENFORCING INVARIANCE IN DIFFUSER

In this section, we show how we may incorporate the three proposed invariance methods from the last section into diffusion models. Enforcing the finite-time invariance in diffusion models is equivalent to ensuring the satisfaction of the conditions in Thms. 3.2-3.4 in the diffusion procedure. In this section, we propose a minimum-deviation quadratic program (QP) approach to achieve that. We wish to enforce these conditions at every step of the diffusion (as shown in Fig. 2) as those states that are far from the specification boundaries $b(\boldsymbol{x}_k^j) = 0$ can also be optimized accordingly, and thus, the model may generate more coherent trajectories.

**Enforcing Invariance for Robust-Safe (RoS) and Time-Varying-Safe Diffusers.** During implementation, the diffusion time step length $\Delta \boldsymbol{\tau}$ in (6) is chosen to be small enough, and we wish the control $\boldsymbol{u}^j$ to stay close to the right-hand side of (6). Thus, we can formulate the following QP-based optimization to find the optimal control for $\boldsymbol{u}^j$ that satisfies the condition in Thms. 3.2 or 3.4:

$$\boldsymbol{u}^{j*} = \arg\min_{\boldsymbol{u}^j} ||\boldsymbol{u}^j - \frac{\boldsymbol{\tau}^j - \boldsymbol{\tau}^{j+1}}{\Delta \boldsymbol{\tau}}||^2, \text{ s.t., (8) if RoS diffuser else s.t., (12)}, \tag{13}$$

where $|| \cdot ||$ denotes the 2-norm of a vector. If we have more than one specification, we can add the corresponding conditions in Thm. 3.2 for each of them to the above QP. After we solve the above

---

**Algorithm 1** Enforcing invariance in diffusion models within a diffusion step

---

**Input:** the last trajectory of diffusion $\boldsymbol{\tau}^{j+1}$ at diffusion step $j \in \{0, \ldots, N\}$
**Output:** safe diffusion state $\boldsymbol{\tau}^{j*}$.
(a) Run diffusion procedure (4) and sample (5) as usual at step $j$ and get $\boldsymbol{\tau}^j$.
(b) Find diffusion dynamics as in (6) - (7).
**if** *Robust-safe diffuser* **then**
    Formulate the QP (13), solve it and get $\boldsymbol{u}^{j*}$.
**else if** Relaxed-safe diffuser **then**
    Define the time-varying weight $w_k(j)$ in (9), formulate the QP (14), solve it and get $\boldsymbol{u}^{j*}, r^{j*}$.
**else**
    Design the time-varying function $\gamma_k(j)$ in (11), formulate the QP (13), solve it and get $\boldsymbol{u}^{j*}$.
**end if**
(c) Update dynamics (7) with $\boldsymbol{u}^j = \boldsymbol{u}^{j*}$ and get $\boldsymbol{\tau}^{j*}$. Finally, $\boldsymbol{\tau}^j \leftarrow \boldsymbol{\tau}^{j*}$.

---

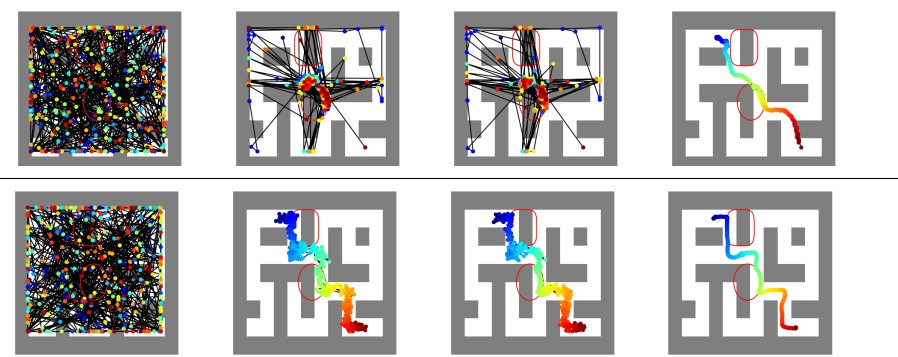

Figure 3: Maze planning (blue to red) denoising diffusion procedure with classifier-based guidance (upper) and time-varying-diffuser (lower). Left to right: diffusion time steps 256, 4, 3, 0, respectively. Red ellipse and super-ellipse denote safe specifications. The classifier-based guidance approach adversely affects the procedure without guarantees. The proposed time-varying safe diffuser can guarantee specifications at the end of diffusion while not significantly affecting the diffusion procedure.

QP and get $\boldsymbol{u}^{j*}$, we update (7) by setting $\boldsymbol{u}^j = \boldsymbol{u}^{j*}$ within the time step and get a new state for the diffusion procedure. Note that all of these happen at the end of each diffusion step.

**Enforcing Invariance for Relaxed-Safe Diffuser.** In this case, since we have relaxation variables for each of the safety specifications, we wish to minimize these relaxations in the cost function to drive all the states towards the satisfaction of specifications. In other words, we have the following QP:

$$\boldsymbol{u}^{j*}, r^{j*} = \arg \min_{\boldsymbol{u}^j, r^j} ||\boldsymbol{u}^j - \frac{\boldsymbol{\tau}^j - \boldsymbol{\tau}^{j+1}}{\Delta \tau}||^2 + ||r^j||^2, \text{ s.t., (10)}, \tag{14}$$

where $r^j$ is the concatenation of $r_k^j$ for all $k \in \{0, \ldots, H\}$. As an alternative, all the constraints above may share the same relaxation variable, i.e., the dimension of $r^j$ is only one. After we solve the above QP and get $\boldsymbol{u}^{j*}$, we update (7) by setting $\boldsymbol{u}^j = \boldsymbol{u}^{j*}$ within the time step and get a new state.

**Complexity of enforcing invariance** The computational complexity of a QP is $\mathscr{O}(q^3)$, where $q$ is the dimension of the decision variable. When there is a set $S$ of specifications, we just add the corresponding constraints for each specification to the QP. The complexity of the three proposed safe diffuser is similar. The algorithm for enforcing invariance is straight forward, which includes the construction of proper conditions, the solving of QP, and the update of diffusion state. We summarize the algorithm in Alg. 1.

Table 1: Maze safe planning comparisons with benchmarks. Items are short for **minimum** metrics of satisfaction of simple specifications (S-SPEC) and complex specifications (C-SPEC), score of planning tasks (SCORE), computation time at each diffusion step (TIME) in seconds, and evidence lower bound (ELBO), respectively. In the method column, items are short for Truncate (Trunc.), Cassifier-guidance (CG), Invariant neural ODE (InvODE), robust-safe diffuser (RoS-DIFFUSER), relaxed-safe diffuser (ReS-DIFFUSER), and time-varying-safe diffuser (TVS-DIFFUSER), relaxed-safe diffuser with last 10 step invariance (ReS-DIFFUSER-l10), respectively. The CG$-\varepsilon$ method applies (safe) gradient when the state is $\varepsilon > 0$ close to the boundary.

| METHOD | S-SPEC($\uparrow$ & $\geq 0$) | C-SPEC($\uparrow$ & $\geq 0$) | SCORE ($\uparrow$) | TIME | ELBO |
|---|---|---|---|---|---|
| DIFFUSER (JANNER ET AL., 2022) | -0.983 | -0.894 | 1.598±0.174 | 0.007 | 4.501±0.475 |
| TRUNC. (BROCKMAN ET AL., 2016) | $-1.192e^{-7}$ | -0.759 | 1.577±0.242 | 0.024 | 4.494±0.465 |
| CG (DHARIWAL & NICHOL, 2021) | -0.789 | -0.979 | 0.384±0.020 | 0.053 | 6.962±0.350 |
| CG$-\varepsilon$ (DHARIWAL & NICHOL, 2021) | -0.853 | -0.995 | 0.383±0.017 | 0.061 | 6.975±0.343 |
| INVODE (XIAO ET AL., 2023B) | 14.000 | $1.657e^{-5}$ | −0.025±0.000 | 0.018 | – |
| RoS-DIFFUSER (OURS) | 0.010 | 0.010 | 1.519±0.330 | 0.106 | 4.584±0.646 |
| ReS-DIFFUSER (OURS) | 0.010 | 0.010 | 1.557±0.289 | 0.107 | 4.434±0.561 |
| TVS-DIFFUSER (OURS) | 0.003 | 0.003 | 1.543±0.303 | 0.107 | 4.533±0.494 |
| ReS-DIFFUSER-L10 (OURS) | 0.010 | 0.010 | 1.527±0.291 | 0.011 | 4.571±0.693 |

## 5 EXPERIMENTS

We set up experiments to answer the following questions: Does our method match the theoretical potential in various tasks quantitatively and qualitatively? How does our method compare with state-of-the-art approaches in enforcing safety specifications? How does our proposed method affect the performance of diffusion under guaranteed specifications? We focus on three experiments from D4RL (Farama-foundation): maze (maze2d-umaze-v1), gym robots (Walker2d-v2 and Hopper-v2), and manipulation. The training data is publicly available, see (Janner et al., 2022). The experiment details and metrics used are shown in Appendix. The safe diffusers generate both planning trajectory and control for the robots, and the score/reward is based on closed-loop control.

### 5.1 SAFE PLANNING IN MAZE

We focus on the case that the training data does not satisfy safety constraints to show how our methods can be generalized to new constraints. For cases where the training data satisfies safety constraints, diffusers may still violate such constraints, while our methods can guarantee safety (see Fig. 9 of Appendix).

The diffuser cannot guarantee the satisfaction of any specifications, as shown in Fig. 1. When using classifier-based guidance in diffusion for safety specifications, the diffusion process could be significantly affected (Fig. 3 upper case). As a result, the generated trajectory will largely deviate from the desired one with no safety. The proposed robust-safe diffuser (RoS-diffuser) may introduce local trap problems (as shown in Fig. 7 of Appendix), but this can be addressed by relaxed-safe diffuser (ReS-diffuser), and time-varying-safe diffuser (TVS-diffuser). The safe diffusers can all guarantee the satisfaction of specifications, even when the specifications are complex (as long as they are differentiable), as shown in Table 1. The proposed methods can also maximally preserve the performance of diffusion models, and this is demonstrated by the scores and evidence lower bound (ELBO) in Table 1, as well as shown by Fig. 3 lower case. The computation time of safe diffusers can be significantly reduced by applying the invariance method to limited diffuser steps, as shown in the last column of Table 1 (0.011s v.s. 0.007s of diffuser). The invariant neural ODE method (Xiao et al., 2023b) can guarantee safety for planning, but it does not work well in the closed-loop control (Fig. 8 of appendix), as shown by the score (-0.025) in Table 1.

### 5.2 SAFE PLANNING FOR ROBOT LOCOMOTION

For robot locomotion, there is no local trap problem, we only consider robust-safe diffuser (RoS-diffuser). Others work similarly. As expected, collisions with the roof are very likely to happen in the walker and hopper using the diffuser since there are no guarantees, as shown in Table 2. The truncation

Table 2: Robot safe planning comparisons with benchmarks. Abbreviations are the same as Table 1.

| EXPERIMENT | METHOD | S-SPEC(↑ & ≥ 0) | C-SPEC(↑ & ≥ 0) | SCORE (↑) | TIME |
|---|---|---|---|---|---|
| | DIFFUSER (JANNER ET AL., 2022) | -9.375 | -4.891 | 0.346±0.106 | 0.037 |
| WALKER2D | TRUNC. (BROCKMAN ET AL., 2016) | 0.0 | × | 0.286±0.180 | 0.105 |
| | CG (DHARIWAL & NICHOL, 2021) | -0.575 | -0.326 | 0.208±0.140 | 0.053 |
| | RoS-DIFFUSER (OURS) | 0.000 | 0.010 | 0.312±0.165 | 0.183 |
| | DIFFUSER (JANNER ET AL., 2022) | -2.180 | -1.862 | 0.455±0.038 | 0.038 |
| HOPPER | TRUNC. (BROCKMAN ET AL., 2016) | 0.0 | × | 0.436±0.067 | 0.046 |
| | CG(DHARIWAL & NICHOL, 2021) | -0.894 | -0.524 | 0.478±0.038 | 0.047 |
| | RoS-DIFFUSER (OURS) | 0.000 | 0.010 | 0.430±0.040 | 0.170 |

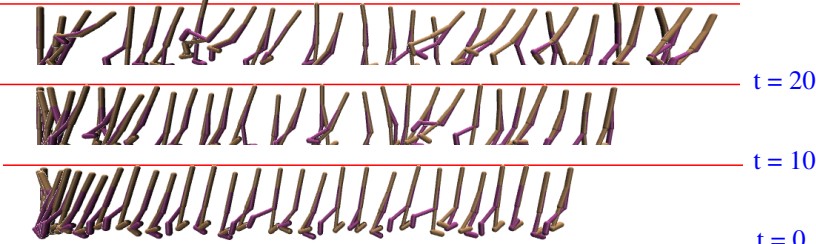

t = 20

t = 10

t = 0

Figure 4: Walker2D planning denoising diffusion with the robust-safe diffuser (Up to down: diffusion time steps 20, 10, 0, respectively). The red line denotes the roof the walker needs to safely avoid during locomotion (safety specifications). Safety is violated at step 20 since the trajectory is initially Gaussian noise, but is eventually guaranteed (step 0). Note that the robot top could touch the roof, and this is not a collision. This can be avoided by defining a tighter safety constraint, i.e., defining a safety constraint $b(x) \geq \sigma$, instead of $b(x) \geq 0$, where $\sigma > 0$ is a small constant.

method can work for simple specifications (S-spec), but not for complex specifications (C-spec). The classifier-based guidance can improve the satisfaction of specifications but with no guarantees. Collision-free is guaranteed using the proposed Ros-diffuser, and one example of diffusion procedure is shown in Fig. 4.

## 5.3 SAFE PLANNING FOR MANIPULATION

Table 3: Manipulation planning comparisons. Abbreviations are the same as Table 1.

| METHOD | S-SPEC(↑ & ≥ 0) | C-SPEC(↑ & ≥ 0) | REWARD (↑) | TIME |
|---|---|---|---|---|
| DIFFUSER (JANNER ET AL., 2022) | -0.057 | -0.065 | 0.650±0.107 | 0.038 |
| TRUNC. (BROCKMAN ET AL., 2016) | $1.631e^{-8}$ | × | 0.575±0.112 | 0.069 |
| CG (DHARIWAL & NICHOL, 2021) | -0.050 | -0.053 | 0.800±0.328 | 0.075 |
| RoS-DIFFUSER (OURS) | 0.072 | 0.069 | 0.925±0.107 | 0.088 |

For manipulation, specifications are joint limitations to avoid collision in joint space. In this case, the truncation method still fails to work for complex specifications (speed-dependent joint limitations). Our proposed RoS-diffuser can work for all specifications as long as they are differentiable. An interesting observation is that the proposed RoS-diffuser can even improve the performance (reward) of diffusion models in this case, as shown in Table 3. This may be due to the fact that the satisfaction of joint limitations can avoid collision in the joint space of the robot as Pybullet is a physical simulator. The computation time of the proposed RoS-diffuser is comparable to other methods. An illustration of the safe diffusion and manipulation procedure is shown in Fig. 5.

## 6 RELATED WORKS

**Diffusion models and planning** Diffusion models (Sohl-Dickstein et al., 2015) (Ho et al., 2020) are data-driven generative modeling tools, widely used in applications to image generations (Dhariwal &

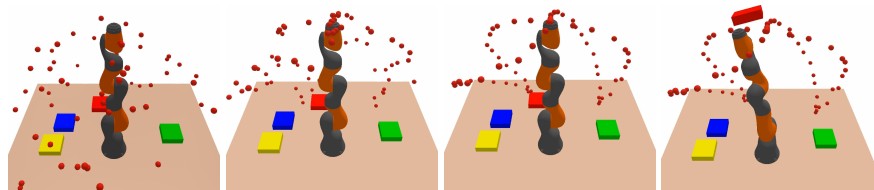

Figure 5: Manipulation planning denoising diffusion procedure with the proposed robust-safe diffuser (Left to right: diffusion time steps 1000, 100, 0, and execution time step 100, respectively). The red dots denote the planning trajectory of the end-effector.

Nichol, 2021) (Du et al., 2020b), in planning (Hafner et al., 2019) (Janner et al., 2021) (Ozair et al., 2021) (Janner et al., 2022), and in language (Saharia et al., 2022) (Liu et al., 2023). Generative models are combined with reinforcement learning to explore dynamic models in the form of convolutional U-networks (Kaiser et al., 2019), stochastic recurrent networks (Ke et al., 2019), neural ODEs (Du et al., 2020a), generative adversarial networks (Eysenbach et al., 2022), neural radiance fields (Li et al., 2022), and transformers (Chen et al., 2022). Further, planning tasks are becoming increasingly important for diffusion models (Lambert et al., 2021) (Ozair et al., 2021) (Janner et al., 2022) as they can generalize well in all kinds of robotic problems. Existing methods for improving the safety of diffusion models employ safety constraints to guide the diffusion process (Yuan et al., 2022) (Ajay et al., 2023). However, there are no methods to equip diffusion models with safety guarantees, which is especially important for safety-critical applications. Here, we address this issue using the proposed finite-time diffusion invariance.

**Set invariance and CBFs.** An invariant set has been widely used to represent the safe behavior of dynamical systems (Preindl, 2016) (Rakovic et al., 2005) (Ames et al., 2017) (Glotfelter et al., 2017) (Xiao & Belta, 2019). In the state of the art of control, Control Barrier Functions (CBFs) are also widely used to prove set invariance (Aubin, 2009), (Prajna et al., 2007), (Wisniewski & Sloth, 2013). CBFs can be traced back to optimization problems (Boyd & Vandenberghe, 2004), and are Lyapunov-like functions (Wieland & Allgöwer, 2007). For time-varying systems, CBFs can also be adapted accordingly (Lindemann & Dimarogonas, 2018). Existing CBF approaches are usually applied in planning time since they are closely coupled with system dynamics. There are few studies of CBFs in other space, such as the diffusion time. Our work addresses all these limitations.

**Guarantees in neural networks.** Differentiable optimization methods show promise for neural network controllers with guarantees (Pereira et al., 2020; Amos et al., 2018; Xiao et al., 2023a). They are usually served as a layer (filter) in the neural networks. In (Amos & Kolter, 2017), a differentiable quadratic program (QP) layer, called OptNet, was introduced. OptNet with CBFs has been used in neural networks as a filter for safe controls (Pereira et al., 2020), in which CBFs are not trainable, thus, potentially limiting the system's learning performance. In (Deshmukh et al., 2019; Zhao et al., 2021; Ferlez et al., 2020), safety guaranteed neural network controllers have been learned through verification-in-the-loop training. The verification approaches cannot ensure coverage of the entire state space. More recently, CBFs have been incorporated into neural ODEs to equip them with specification guarantees (Xiao et al., 2023b). However, none of these methods can be applied in diffusion models, which we address in this paper.

## 7    CONCLUSIONS, LIMITATIONS AND FUTURE WORK

We have proposed finite-time diffusion invariance for diffusion models to ensure safe planning. We have demonstrated the effectiveness of our method on a series of robotic planning tasks. Nonetheless, our method faces a few shortcomings motivating for future work.

**Limitations.** Specifically, specifications for diffusion models should be expressed as continuously differentiable constraints that may be unknown for planning tasks. Further work may explore how to learn specifications from history trajectory data. The computation time is much higher than the diffuser if we apply invariance to every diffusion step. This can be improved by applying invariance to a limited number of diffusion steps. Moreover, there is also a gap between planning and control using diffusion models. We may further investigate diffusion for safe control policies when robot dynamics are known or to be learned.

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

## A PROOF

### A.1 PROOF OF THM. 3.2

**Proof:** Given a continuously differentiable constraint $h(x_t) \geq 0$ ($h(x_0) \geq 0$), by Nagumo's theorem (Nagumo, 1942), the necessary and sufficient condition for the satisfaction of $h(x_t) \geq 0, \forall t \geq 0$ is

$$\dot{h}(x_t) \geq 0, \text{ when } h(x_t) = 0,$$

If $b(x_k^N) - \gamma(N, \varepsilon) \geq 0, k \in \{0, \ldots, H\}$, then the condition (8) is equivalent to

$$\frac{db(x_k^j)}{dx_k^j} \dot{x}_k^j + \varepsilon(b(x_k^j) - \gamma(N, \varepsilon)) \geq 0,$$

where $\dot{x}_k^j$ is the diffusion time derivative. The last equation is equivalent to

$$\frac{d(b(x_k^j) - \gamma(N, \varepsilon))}{dT} + \varepsilon(b(x_k^j) - \gamma(N, \varepsilon)) \geq 0,$$

where $T$ denotes the diffusion time.

Further, we have that

$$\varepsilon(b(x_k^j) - \gamma(N, \varepsilon)) \to 0, \text{ as } b(x_k^j) \to \gamma(N, \varepsilon),$$

In other words, we have $\frac{d(b(x_k^j) - \gamma(N, \varepsilon))}{dT} \geq 0$ when $b(x_k^j) = \gamma(N, \varepsilon)$. Since $b(x_k^N) \geq \gamma(N, \varepsilon), k \in \{0, \ldots, H\}$, then by Nagumo's theorem, we have $b(x_k^j) \geq \gamma(N, \varepsilon) > 0, \forall j \in \{0, \ldots, N-1\}$. Therefore, the diffusion procedure $p_\theta(\tau^{i-1}|\tau^i), i \in \{1, \ldots, N\}$ is finite-time diffusion invariant, and the finite time in diffusion invariance is $N$.

If, on the other hand, $b(x_k^N) < \gamma(N, \varepsilon), k \in \{0, \ldots, H\}$, then we can define a Lyapunov function:

$$V(x_k^j) = \gamma(N, \varepsilon) - b(x_k^j), k \in \{0, \ldots, H\}, j \in \{0, \ldots, N\}, \tag{15}$$

and $V(x_k^N) > 0$.

Replacing $b(x_k^j) - \gamma(N, \varepsilon)$ by $V(x_k^j)$, the condition (8) is equivalent to (note that $\dot{x}_k^j = u_k^j$)

$$\frac{dV(x_k^j)}{dx_k^j} \dot{x}_k^j + \varepsilon V(x_k^j) \leq 0,$$

which is equivalent to

$$\dot{V}(\boldsymbol{x}_k^j) + \varepsilon V(\boldsymbol{x}_k^j) \leq 0,$$

Suppose we have

$$\dot{V}(\boldsymbol{x}_k^j) + \varepsilon V(\boldsymbol{x}_k^j) = 0,$$

the solution to the above equation is

$$V(\boldsymbol{x}_k^j) = V(\boldsymbol{x}_k^N)e^{-\varepsilon(N-j)},$$

Using the comparison lemma (Khalil, 2002), equation (8) implies that

$$V(\boldsymbol{x}_k^j) \leq V(\boldsymbol{x}_k^N)e^{-\varepsilon(N-j)}, j \in \{0,\ldots,N\},$$

At diffusion step 0, i.e., $j = 0$, the last inequality becomes

$$V(\boldsymbol{x}_k) \leq V(\boldsymbol{x}_k^N)e^{-\varepsilon N}, k \in \{0,\ldots,H\},$$

Substituting $V(\boldsymbol{x}_k^j) = \gamma(N,\varepsilon) - b(\boldsymbol{x}_k^j), j \in \{0,\ldots,N\}$ into the last equation, we have

$$\gamma(N,\varepsilon) - b(\boldsymbol{x}_k) \leq (\gamma(N,\varepsilon) - b(\boldsymbol{x}_k^N))e^{-\varepsilon N}, k \in \{0,\ldots,H\},$$

Since $b(\boldsymbol{x}_k^N) < \gamma(N,\varepsilon)$ in this case, the last equation can be rewritten as

$$-b(\boldsymbol{x}_k) \leq |\gamma(N,\varepsilon) - b(\boldsymbol{x}_k^N)|e^{-\varepsilon N} - \gamma(N,\varepsilon), k \in \{0,\ldots,H\},$$

Following the condition $\gamma(N,\varepsilon) \geq |\gamma(N,\varepsilon) - b(x_k^N)|e^{-\varepsilon N}$ in the theorem, we have

$$-b(\boldsymbol{x}_k) \leq |\gamma(N,\varepsilon) - b(\boldsymbol{x}_k^N)|e^{-\varepsilon N} - \gamma(N,\varepsilon) \leq 0, k \in \{0,\ldots,H\},$$

Therefore,

$$b(\boldsymbol{x}_k) \geq 0, \forall k \in \{0,\ldots,H\},$$

the diffusion procedure $p_\theta(\boldsymbol{\tau}^{i-1}|\boldsymbol{\tau}^i), i \in \{1,\ldots,N\}$ is finite-time diffusion invariant. ∎

## A.2 PROOF OF THM. 3.3

**Proof:** Since the weight $w_k(j)$ is chosen such that $w_k(j) = 0$ for all $j \leq N_0, 0 \leq N_0 \leq N-1$, then the condition (10) becomes a hard constraint when $j < N_0$. In other words, equation (10) becomes:

$$h(\boldsymbol{u}_k^j|\boldsymbol{x}_k^j) := \frac{db(\boldsymbol{x}_k^j)}{d\boldsymbol{x}_k^j}\boldsymbol{u}_k^j + \alpha(b(\boldsymbol{x}_k^j)) \geq 0, k \in \{0,\ldots,H\}, j \in \{0,\ldots,N_0\},$$

Then, the proof is similar to that of the Thm. 3.2, and we have that the diffusion procedure $p_\theta(\boldsymbol{\tau}^{i-1}|\boldsymbol{\tau}^i), i \in \{0,\ldots,N\}$ is finite-time diffusion invariant. ∎

## A.3 PROOF OF THM. 3.4

**Proof:** Since $\gamma_k(N) \leq b(\boldsymbol{x}_k^N)$, we have that $s(\boldsymbol{x}_k^j, \gamma_k(j)) := b(\boldsymbol{x}_k^j) - \gamma_k(j) \geq 0$ when $j = N$.

The condition (12) is equivalent to

$$\frac{\partial s(\boldsymbol{x}_k^j, \gamma_k(j))}{\partial \boldsymbol{x}_k^j}\boldsymbol{u}_k^j + \frac{\partial s(\boldsymbol{x}_k^j, \gamma_k(j))}{\partial j} + \alpha(s(\boldsymbol{x}_k^j, \gamma_k(j))) \geq 0,$$

which can be rewritten as

$$\dot{s}(\boldsymbol{x}_k^j, \gamma_k(j)) + \alpha(s(\boldsymbol{x}_k^j, \gamma_k(j))) \geq 0,$$

Using the Nagumo' theorem presented in the proof of Thm. 3.2, we have that

$$s(\boldsymbol{x}_k^j, \gamma_k(j)) \geq 0, \forall j \in \{0,\ldots,N\}$$

since $s(\boldsymbol{x}_k^N, \gamma_k(N)) \geq 0$.

As $\gamma_k(0) = 0$ and $s(\boldsymbol{x}_k^j, \gamma_k(j)) := b(\boldsymbol{x}_k^j) - \gamma_k(j)$, we have that $b(\boldsymbol{x}_k^0) \geq 0, \forall k \in \{0,\ldots,H\}$. Therefore, the diffusion procedure $p_\theta(\boldsymbol{\tau}^{i-1}|\boldsymbol{\tau}^i), i \in \{0,\ldots,N\}$ is finite-time diffusion invariant, and the finite time in diffusion invariance is 0. ∎

## B    EXPERIMENT DETAILS

**Planning Tasks (Farama-foundation/d4rl/wiki/Tasks)**

**Maze.** In maze planning, we aim to impose trajectory constraints on the planning path of a maze. The initial positions and destinations in maze are randomly generated. The diffusion model is conditioned on the initial positions and destinations.

**Robot locomotion.** For robot locomotion (in MuJoCo), we wish the robot to avoid collisions with obstacles, such as the roof. In this case, since there is no local trap problem, we only consider robust-safe diffuser (RoS-diffuser). Others work similarly.

**Manipulation.** For manipulation (in Pybullet), the diffusion models generate joint trajectories (as controls) for the robot, which are conditioned on the locations of the objects to grasp and place. Specifications are joint limitations to avoid collision in joint space.

**Metrics and methods used in Tables**

**Classifier-based guidance** is done by applying gradients (towards safe space) to the trajectory points that drive them to the safe side of the space whenever safety constraints are violated.

The **score** (results from closed-loop control) represents the normalized reward that is used in RL. The problem uses a sparse reward which has a value of 1.0 when the agent is within a 0.5 unit radius of the target. Specifically, we use the env.get_normalized_score(returns) function in dr4l to compute a normalized score for an episode, where returns are the undiscounted total sum of rewards accumulated during an episode.

**S-spec and C-spec.** Simple specifications (S-spec) and Complex specifications (C-spec) are calculated by the minimum values of the functions (e.g., $b(x)$) among all runs that define the safety constraints (e.g., $b(x) \geq 0$). They are defined by how complex the specifications are. For instance, in the maze example, the elliptical obstacle is defined as a simple obstacle as we can easily apply a truncation method to satisfy the safety constraints, while the supper-elliptical obstacle is defined as a complex obstacle as it is hard to apply the truncation method.

The **ELBO** for all the baselines and our methods follow the function in guided-diffusion/guided_diffusion/gaussian_diffusion.GaussianDiffusion._vb_terms_bpd from (Dhariwal & Nichol, 2021).

### B.1    SAFE PLANNING IN MAZE

In this experiment, we aim to impose trajectory constraints on the planning path of a maze. The training data is publicly available from (Janner et al., 2022), in which initial positions and destinations in maze are randomly generated. The diffusion model is conditioned on the initial positions and destinations.

**Specifications.** The simple safety specification for the planning trajectory is defined as a super-ellipse-shape obstacle:

$$\left(\frac{x-x_0}{a}\right)^2 + \left(\frac{y-y_0}{b}\right)^2 \geq 1, \tag{16}$$

where $(x,y) \in \mathbb{R}^2$ is the state on the planning trajectory, $(x_0, y_0) \in \mathbb{R}^2$ is the location of the obstacle. $a > 0, b > 0$. Since the state $(x,y)$ is normalized in diffusion models, we also need to normalize the above constraint accordingly. In other words, we normalize $x_0, a$ and $y_0, b$ according to the normalization of $(x,y)$ along the $x$-axis and $y-$axis, respectively.

The complex safety specification for the planning trajectory is defined as an ellipse-shape obstacle:

$$\left(\frac{x-x_0}{a}\right)^4 + \left(\frac{y-y_0}{b}\right)^4 \geq 1, \tag{17}$$

We also normalize the above constraint as in the simple case. In this case, it is non-trivial to truncate the planning trajectory to satisfy the constraint. When we have much more complex specifications, it is too hard for the truncation method to work.

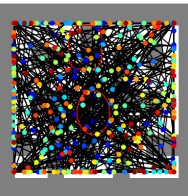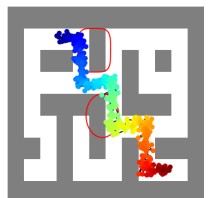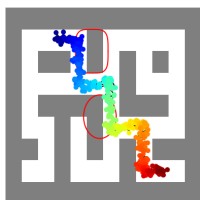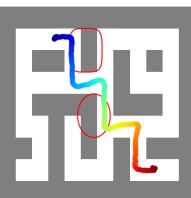

Figure 6: Maze planning (blue to red) denoising diffusion procedure with diffuser (Left to right: diffusion time steps 256, 4, 3, 0, respectively). Red ellipse and superellipse (outside) denote safe specifications. Both specifications are violated with the trajectory from diffuser.

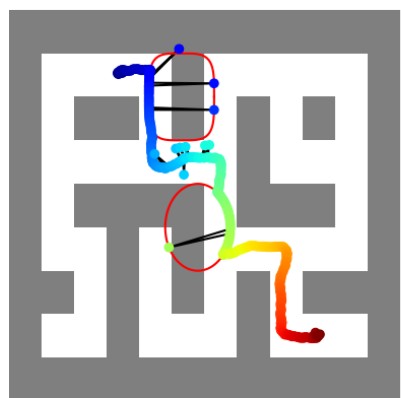

Figure 7: Maze planning (blue to red) denoising diffusion procedure with robust-safe diffuser at diffusion time step 0. Red ellipse and superellipse (outside) denote safe specifications. Although with safety guarantees, some trajectory points may get stuck in local traps.

**Model setup, training and testing.** The diffusion model structure is the same as the open source one (Maze2D-large-v1) provided in (Janner et al., 2022). We set the planning horizon as 384, the diffusion steps as 256 for the proposed methods. The learning rate is $2e^{-4}$ with $2e^{6}$ training steps. The training of the model takes about 10 hours on a Nvidia RTX-3090 GPU. More parameters are provided in the attached code: "safediffuser/config/maze2d.py". The switch of different (proposed) methods in testing can be modified in "safediffuser/diffuser/models/diffusion.py" through "GaussianDiffusion.p_sample()" function.

In Fig. 6, we present a diffusion procedure using the diffuser, in which case the generated trajectory can easily violate safety constraints. Using the proposed robust-safe diffuser, the generated trajectory can guarantee safety, but some points on the trajectory may get stuck in local traps, as shown in 7. Using the proposed relaxed-safe diffuser and time-varying-safe diffuser, the local trap problem could be addressed.

**invariant neural ODE.** The invariant neural ODE method (Xiao et al., 2023b) does not work well in closed-loop control, as shown in Fig. 8.

**Maze2d-umaze-v1 (training data satisfies safety constraints).** In this case, the training data satisfies the safety constraints (modeled as maze walls). The diffuser may still violate such safety constraints due to uncertainties in inference, as shown in Fig. 9 left case. While our safe diffuser can guarantee safety (Initial positions and destinations are randomly generated).

**Comparison between one-step safe diffusers and multi-step safe diffusers.** The safe diffusers have to be applied for multiple diffusion steps, otherwise, the safety constraints may still be violated (as shown in Fig. 10 left case) as the proposed diffusion invariance is a dynamic process that requires several iterations to drive the trajectory to safe space.

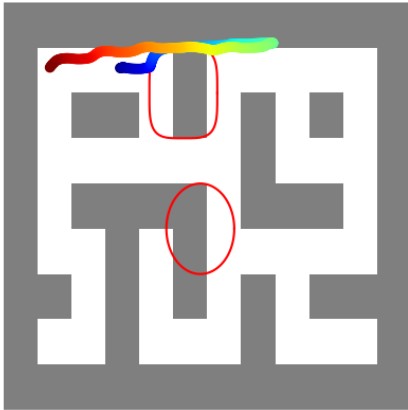

Figure 8: Maze planning (blue to red) using neural ODEs with invariance. Although with safety guarantees, the neural ODEs fail to work for such a long horizon planning problem.

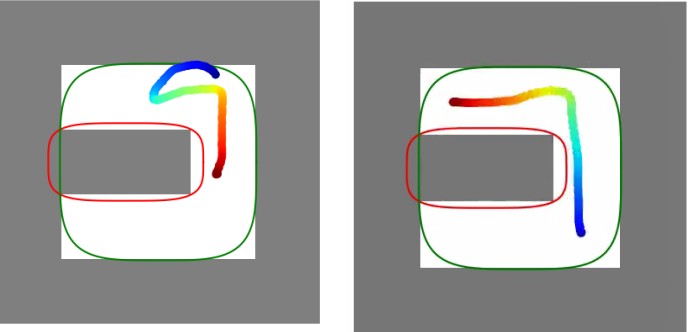

Figure 9: Maze planning (blue to red) denoising diffusion procedure using diffuser (left) and safediffuser (right) when the training data satisfies safety constraints. Red and green super-ellipses denote safe specifications for the walls. Diffusers may still violate safety constraints.

**Safety portrait statistics.** We present in Fig. 11 the distribution of safety portrait (C-spec) among 100 runs with respect to scores. In summary, our safe diffusers can guarantee safety while maximally preserving performance.

### B.2 Safe Planning for Robot Locomotion

For robot locomotion (in MuJoCo), we wish the robot to avoid collisions with obstacles, such as the roof. In this case, since there is no local trap problem, we only consider robust-safe diffuser (RoS-diffuser). Others work similarly. The training data set is publicly available from (Janner et al., 2022).

**Specifications.** The simple safety specification for both the Walker2D and Hopper is collision avoidance with the roof. In other words, the height of the robot head $z \in \mathbb{R}$ should satisfy the following constraint:

$$z \leq h_r, \tag{18}$$

where $h_r > 0$ is the height of the roof. We also need to normalize $h_r$ according to the normalization of the state $z$ in the diffusion model.

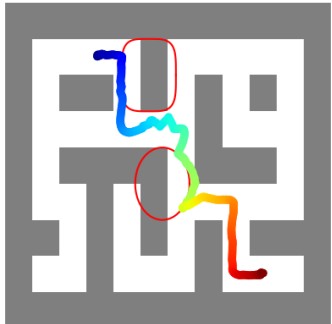 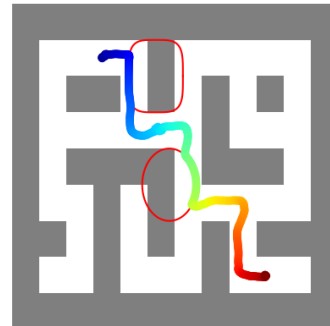

Figure 10: Maze planning (blue to red) denoising diffusion procedure using safediffuser for the last step (left) and the last 10 steps (right). The generated trajectory is more smooth with safety guarantees when running for the last 10 steps.

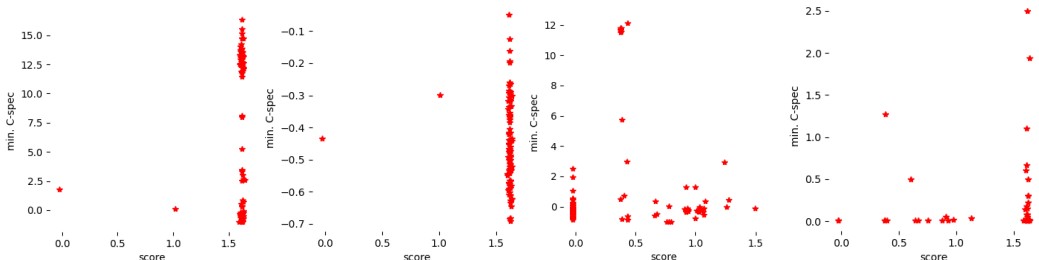

Figure 11: Specification satisfaction metrics (C-spec) v.s. scores in maze planning using diffusion models (from left to right: diffuser, truncation, classifier guidance, safediffuser). Safediffusers can guarantee the satisfaction of specifications while making the scores consistently close to 1.6 among all the runs (mean value: 1.527).

The complex safety specification for both the Walker2D and Hopper is a speed-dependent collision avoidance constraint:

$$z + \varphi v_z \leq h_r, \tag{19}$$

where $\varphi > 0$, $v_z \in \mathbb{R}$ is the speed of the robot head along the $z$-axis. The speed-dependent safety constraint is more robust for the robot to avoid collision with the roof since when the robot jumps faster, we need to ensure a larger safe distance with respect to the roof in order to account for all kinds of uncertainties or perturbations. In this case, the simple truncation method is hard to work since it is not clear how to truncate both $z$ and $v_z$ at the same time.

**Model setup, training and testing.** The diffusion model structures are the same as the open source ones (Walker2D-Medium-Expert-v2 and Hopper-Medium-Expert-v2) provided in (Janner et al., 2022). We set the planning horizon as 600, the diffusion steps as 20. The learning rate is $2e^{-4}$ with $2e^6$ training steps. The training of the model takes about 16 hours on a Nvidia RTX-3090 GPU. More parameters are provided in the attached code: "safediffuser/config/locomotion.py". The switch of different methods in testing can be modified in "safediffuser/diffuser/models/diffusion.py" through "GaussianDiffusion.p_sample()" function.

### B.3 SAFE PLANNING FOR MANIPULATION

For manipulation (in Pybullet), the diffusion models generate joint trajectories (as controls) for the robot, which are conditioned on the locations of the objects to grasp and place. The training data set is publicly available from (Janner et al., 2022). Specifications are joint limitations to avoid collision in joint space.

**Specifications.** The simple safety specification for the robot is in the joint space, and we are trying to limit the joint angles of the robot within allowed ranges:

$$x_{min} \le x \le x_{max}, \tag{20}$$

where $x \in \mathbb{R}^7$ is the state of 7 joint angles, $x_{min} \in \mathbb{R}^7$ and $x_{max} \in \mathbb{R}^7$ denotes the minimum and maximum joint limits. We need to normalize the limits according to how the state $x$ is normalized in the diffusion model.

The complex safety specifications are speed-dependent joint constraints:

$$x_{min} \le x + \varphi v \le x_{max}, \tag{21}$$

where $\varphi > 0$, $v \in \mathbb{R}^7$ is the joint speed corresponding to the joint angle $x$. In this example, since the diffusion model does not directly predict $v$, we evaluate $v$ using $x(k)$ and $x(k+1)$ along the planning horizon. The joints limits are also normalized as in the simple specification case.

**Model setup, training and testing.** The diffusion model structure is the same as the open source one provided in (Janner et al., 2022), and we use their pre-trained models to evaluate our methods when comparing with other approaches.

