# OpenReview forum: "SafeDiffuser: Safe Planning with Diffusion Probabilistic Models"
_ICLR.cc/2024/Conference — ICLR 2024 Conference Withdrawn Submission_

### Official Review · Reviewer_dMwB · 2023-10-28

**Soundness:** 2 fair
**Presentation:** 3 good
**Contribution:** 3 good
**Rating:** 6
**Confidence:** 3

**Summary:**

Diffusion models have been a successful generative modelling mechanism in image synthesis. Many works have made creative extensions to use diffusion models for planning and trajectory synthesis. Most of the previous work doesn’t explicitly guide the diffusion process to respect hard safety constraints. Inspired by CBF’s and classical theorems known about barrier functions, this paper proposes to project the diffusion dynamics to improve safety constraints over the course of the diffusion process. Three formulations: robust safe diffuser, time-varying safe diffuser and relaxed safe diffuser are defined with the latter two fixing a critical issue with the first approach. The foundational idea of the contribution has mathematical backing and it is possible to prove from Nagumo's theorem that every point of the trajectory will eventually satisfy constraints. Experiments are performed on maze solving, robot locomotion and manipulation to demonstrate and compare with previous baselines.

**Strengths:**

Creative, impactful and sound contribution. As a result, I am leaning towards an acceptance score.
Additionally, I found the extensions of the Robust Safe-Diffuser to time-varying and relaxed Safe-Diffuser to be quite interesting. It is believable that the most basic formulation would get stuck in local traps as it is never allowed to violate safety. Additional deeper insights on this phenomenon are appreciated.

**Weaknesses:**

1) For the experimental comparisons, a detailed description of all the baselines would be useful.

2) I have some questions about the sufficiency of the baselines and comparisons. See below.

3) Would be nice to have a result on one more setup but it is not a major drawback.

**Questions:**

1) https://arxiv.org/pdf/2211.15657.pdf For example here, a one hot encoding is used to guide the diffusion process to satisfy constraints. Is it possible to compare to this? This is different from classifier-based guidance as we are conditioning on a random variable encoding constraint satisfaction.

2) https://arxiv.org/pdf/2205.09991.pdf Is it valid to compare against using diffusion guided by reward or goal-based RL and just use an usual off-the shelf CBF at runtime?

3) With standard RL, there can be one performance reward and another safety reward similar to what is usually done in safe constrained RL. Can something like that be amenable to the diffusion process?

4) Would it be necessary to have a large number of diffusion steps in some cases?

5) Is there a possibility of trajectories not being consistent with the dynamics of the environment? i.e. the generated trajectories can have several violations of the underlying dynamics of the robot. If yes, are there some metrics quantifying the dynamics validity of trajectories.

---

> ### Author Response · Authors · 2023-11-21
> **Response to Reviewer dMwB**
>
> We really appreciate the reviewer for the positive and constructive feedback, and for recognizing the values of our work. Belows are our responses to the reviewer's questions.
>
> **Reviewer weakness 1:** For the experimental comparisons, a detailed description of all the baselines would be useful.
>
> **Response:** We would add more description of all the baselines in the revision.
>
>
> **Reviewer question 1:** https://arxiv.org/pdf/2211.15657.pdf For example here, a one hot encoding is used to guide the diffusion process to satisfy constraints. Is it possible to compare to this? This is different from classifier-based guidance as we are conditioning on a random variable encoding constraint satisfaction.
>
> **Response:** This is the decision diffuser that we cited as [1] in our paper. We agree that it would be nice to have it as a baseline. One drawback similar to classifier-based guidance is that the decision diffuser still has no safety guarantees. Thus, we have not included it in our baselines.
>
> **Reviewer question 2:** https://arxiv.org/pdf/2205.09991.pdf Is it valid to compare against using diffusion guided by reward or goal-based RL and just use an usual off-the shelf CBF at runtime?
>
> **Response:** This is actually not a fair comparison. Our method focuses on safe planning, while the off-the-shelf CBF at runtime is about safe control that is implemented at a lower level than safe planning. This off-the-shelf method is actually the shielding method that is popularly used in ML.  When the diffusion process is done, the (diffusion) dynamics of each of the planning trajectory points do not exist anymore, and thus, we cannot use the CBF method to modify the planning trajectories anymore (since the CBF method is based on dynamics).
>
> **Reviewer question 3:** With standard RL, there can be one performance reward and another safety reward similar to what is usually done in safe constrained RL. Can something like that be amenable to the diffusion process?
>
> **Response:** This is possible in the diffusion process, but without any safety guarantees. A safety reward is similar to a soft safety constraint in optimizations, while the CBF method we used can strictly guarantee the safety constraints.. Therefore, we prefer the use of CBFs. Moreover, we focus on offline settings, thus with no access to online interaction with the environment like RL. The closest baselines can be (1) training with safe data and (2) enforce post-hoc safety like shielding, which are both included as baselines in our paper.
>
> **Reviewer question 4:** Would it be necessary to have a large number of diffusion steps in some cases?
>
> **Response:** The number of diffusion steps depends on the applications as it may affect the performance. We can always guarantee safety by choosing a proper robust term $\gamma$ in Thm. 3.2 according to the maximum diffusion step $N$, and the CBF parameter $\varepsilon$.
>
> **Reviewer question 5:** Is there a possibility of trajectories not being consistent with the dynamics of the environment? i.e. the generated trajectories can have several violations of the underlying dynamics of the robot. If yes, are there some metrics quantifying the dynamics validity of trajectories.
>
> **Response:** This is definitely very likely. We have to use a safe controller to account for the dynamics of the environment/robots, which is at a lower level of safe planning. However, we focus on safe planning in this work. We are not considering how these planning trajectories can be safely tracked using a low level controller that complies with the robot dynamics. Nonetheless,  quantifying the dynamics validity of trajectories would be a very interesting future direction.

---

> > ### Comment · Reviewer_dMwB · 2023-11-21
> > **Acknowledgement**
> >
> > I found the paper creative and interesting while reviewing. After reading the response, I am mostly satisfied. However, if the method is restricted to safe planning only and the planner's trajectories are not dynamically consistent, that is a moderate weakness.  Experimental comparisons and ablation studies could be improved. Other reviewers have brought up some serious issues with the paper. Depending on those issues, I choose to revise my scores.

---

### Official Review · Reviewer_HJX4 · 2023-10-30

**Soundness:** 1 poor
**Presentation:** 1 poor
**Contribution:** 2 fair
**Rating:** 3
**Confidence:** 2

**Summary:**

This paper presents a safe planning algorithm using a diffuser planner. The authors incorporate the safety constraints into planning, which is a preferable solution to the problem. The planning algorithm can potentially suffer from the "local traps" - the phenomenon inherited from the planner, however, authors propose a solution to this problem by relaxing the safety constraints. The algorithm is demonstrated on the maze and robot locomotion tasks.

**Strengths:**

1. The authors present theoretical results regarding their method
1. The authors develop the theory of safe diffusion planners
1. They demonstrate the behavior of the safe algorithm on the maze and the robot locomotion tasks.

**Weaknesses:**

1. Some of the theoretical developments and derivations are confusing. For example, the control barrier functions are developed in continuous time, while planning is done in discrete time.
1. There are some issues with notation. For example, it appears that $u$ denotes a control signal for the continuous time systems as well as the diffusion process.
1. It is not clear how the dynamics / one-step transitions are used in the planner. Perhaps I missed it, but I couldn't understand it at all.

Overall the methodology is not clear to me, but perhaps authors can clear this up

**Questions:**

1. Why continuous dynamics and CBF for continuous if in practice it is discrete? This introduces extra estimation errors.
1. The description of the local trap problem is slightly confusing for me. Could the authors elaborate on what is happening in more detail? Is the problem inherent to the diffusion planner?

---

> ### Author Response · Authors · 2023-11-21
> **Response to Reviewer HJX4**
>
> We thank the reviewer for all the feedback regarding clarifications. We will improve it in our revision. Belows are our responses to address all the reviewer's concerns.
>
> **Reviewer weakness 1:** Some of the theoretical developments and derivations are confusing. For example, the control barrier functions are developed in continuous time, while planning is done in discrete time.
>
> **Response:** Any method has to be implemented in discrete time in real applications. The CBF method is indeed developed in continuous time, but there are methods (Taylor, et.al. CDC 2022; Xiao, et.al. TAC2022) to deal with inter-sampling effect due to time discretization. There is also the discrete-time version (Agrawal, et.al., RSS 2017) of the CBF method, but we are not using it since the continuous version is more general as we may have high order CBFs  (Nguyen, et.al., ACC2016;  Xiao, et.al. CDC 2019) and the discrete method tends to be more conservative (Agrawal, et.al., RSS 2017).
>
> **Reviewer weakness 2:** There are some issues with notation. For example, it appears that $u$ denotes a control signal for the continuous time systems as well as the diffusion process.
>
> **Response:** We are actually ``controlling’’ the diffusion process to make it satisfy safety constraints in our methods, and this is a similar concept to the control in continuous time systems. We will change this to some other variable in the revision.
>
> **Reviewer weakness 3:** It is not clear how the dynamics / one-step transitions are used in the planner. Perhaps I missed it, but I couldn't understand it at all.
>
> **Response:** This is summarized in Algorithm 1. In summary, we find the optimal control $u$ for each diffusion time step by solving the QP (13) or (14), and then update the diffusion dynamics (7) with the $u$ to get the next diffusion state variable (i.e., planning trajectory points at each diffusion step). This is a general application method based on a recursive sample, control, and update process  for control systems (Nguyen, et.al., ACC2016; Taylor, et.al. CDC 2022).
>
> **Reviewer weakness summary:** Overall the methodology is not clear to me, but perhaps authors can clear this up
>
>
> **Response:** Algorithm 1 summarizes our methodology. We would improve the description in the revision.
>
>
>
>
> **Reviewer question 1:** Why continuous dynamics and CBF for continuous if in practice it is discrete? This introduces extra estimation errors.
>
> **Response:** Continuous time-version CBFs are more general and less conservative, and there are existing methods (Taylor, et.al. CDC 2022; Xiao, et.al. TAC2022) in CBFs to address the estimation errors (i.e., address the inter-sampling effect).
>
> **Reviewer question 2:** The description of the local trap problem is slightly confusing for me. Could the authors elaborate on what is happening in more detail? Is the problem inherent to the diffusion planner?
>
> **Response:** This is actually due to the enforcement of hard safety constraints. When there are non-convex obstacles/linear boundaries, the trajectory points in the diffusion process could easily get stuck at these locations. This is what we refer to as the local trap problem. This is not inherent to the diffusion planner unless we have hard safety constraints for the planning trajectory. See fig. 7 for an example. There are some trajectory points getting stuck at the local trap at the end of the diffusion process.

---

> > ### Comment · Reviewer_HJX4 · 2023-11-21
> > **Thank you for the response!**
> >
> > I want to thank the authors for the response, however, the response hasn't clarified my questions and the manuscript doesn't seem to be updated.
> >
> > Regarding continuous/discrete dynamics. The higher-order CBFs are not used in the paper and the control affine assumption (the only reason to use continuous-time over discrete) doesn't seem to be needed. Although again I am not sure how the dynamics are used. I appreciate that this is a design choice, but it needed to be discussed in the paper as it is a non-standard choice.
> >
> > I maintain my score.

---

### Official Review · Reviewer_3To6 · 2023-10-31

**Soundness:** 2 fair
**Presentation:** 1 poor
**Contribution:** 2 fair
**Rating:** 1
**Confidence:** 3

**Summary:**

This paper recognizes the importance of establishing safety guarantee in the application of diffusion process planning, and introduces three safe diffusers from the perspective of control theory and control barrier functions. Particularly, this paper attempts to address the local traps during diffusion procedure.

**Strengths:**

1. It's an interesting perspective to work around with finite-time diffusion invariance to guarantee safety constraint in the diffusion models.
2. Propose three safe diffusers and work through the theorems related to each of them respectively and demonstrate they achieve decent results in the experiment section.

**Weaknesses:**

1. Experimental design is not clear. The details of how time-varying weight $w_k$ or time varying function $\gamma_k$ are not revealed. I believe they have to be carefully chosen to guarantee the effect of the safe diffusers and ablation studies are needed here. More ablation studies are needed when determine number of diffusion time steps to reduce the computation time while still maintaining the model performance.
2. (a) Some typos in the text: e.g., 'Cassifier-guidance' in the description of Table 1. (b) The ``ELBO'' in the tables should be negative. (c) Some typos in the theorems proof: e.g., proof of theorem 3.2, at the very bottom of page 12, "Replace $b(x_k^j) - \gamma(N, \epsilon)$ by $V(x_k^j)$'', $b$ and $\gamma$ should be switched over. (d) In theorem 3.2, All equations and inequalities should have labels. Some equations and statements should have period at the end but not comma everywhere.

**Questions:**

1. The figure 3 compares the samples between classifier-based guidance and time-varying-diffuser. It is hard to believe that the classifier-based guidance is much worse than the time-varying diffuser at diffusion time steps 4 and 3. Do you have some idea why it stays so noisy at the very end of denoting steps but become noiseless at time step 0 (ignore the constraint violation)? Also, why classifier-base guidance failed to identify the constraint and steer the trajectory away from the obstacles?
2. I have a hard time to understand what definition 2.3 Control Barrier Function really means and tries to delivery. Can you give an intuitive explanation for it? Or some geometric meaning?
3. In the proof of theorem 3.2, I don't really understand why $\frac{d(b(x_k^j) - \gamma(N, \epsilon))}{dt} + \epsilon(b(x_k^j) - \gamma(N, \epsilon))\geq 0$ (the third inequality) ``is equivalent to the last equation (inequality)''. What is exactly diffusion time $T$ here? I know $\gamma$ is a robust term, but what's the justification of its appearance?
4. In the experiments, what are their corresponding robust function $\gamma(\cdot)$, extended class function $\alpha(\cdot)$? Can you revel more details about them? What does $r$ represent in equation 14?

**Details Of Ethics Concerns:**

The subsections Diffusion Probabilistic Models and Notations in sections 2 Preliminaries have significant similarities to subsection 2.2 in section 2 Background from the paper ``Planning with Diffusion for Flexible Behavior Synthesis'' [1], and this may indicate a potential plagiarism.

[1] Janner, Michael, et al. "Planning with diffusion for flexible behavior synthesis." arXiv preprint arXiv:2205.09991 (2022).

---

> ### Author Response · Authors · 2023-11-21
> **Response to Reviewer 3To6**
>
> We appreciate the reviewer for all the constructive comments, especially the ethics review. We are aware of this problem, and have rewritten the subsection regarding diffusion models to avoid such problems. Please see the new revised draft.
>
> **Reviewer weakness 1:** Experimental design is not clear. ...
>
> **Response:** $w_k(j)$ is defined as $w_k(j) = 100$ when $j \geq 10$; otherwise,  $w_k(j) = 0$ in the maze example. $\gamma_k(j) = sigmoid(j_{bias} - j)$, where $j_{bias}$ = 5, in which case $\gamma_k(N)$ is near 0 at the beginning of the diffusion time $j = N$. Therefore, the unsafe set is very small such that all the trajectory points are outside the unsafe set. When $j = 0$, $\gamma_k(j)$ is close to 1, and the safety constraint is satisfied for all the trajectory points  (note that the safety constraint is define as $((x_k - o_x)/x_r)^2 + ((y_k - o_y)/y_r)^2 - 1\geq 0$, where $(o_x, o_y)$ is the 2D location of the ellipse, and $x_r, y_r$ denote the half lengths of the ellipsoid-shape obstacle. $(x_k. y_k)$ is the 2D location of the trajectory point $k$).  There are many different ways to define both time-varying functions, and their definitions do not greatly affect the safety satisfaction as long as the initial (i.e., the unsafe set is very small such that all the trajectory points are initially outside the unsafe set) and terminal conditions (the safe set is the same as the safety constraint specification) are satisfied.
>
> We have done ablation studies on the number of diffusion time steps to reduce the computation time while still maintaining the model performance. It actually only requires 3 time steps (this is set to 10 in Table I to ensure robustness) for the maze example as the trajectory points can quickly converge to the safe set.
>
>
> **Reviewer weakness 2:** (a) Some typos in the text: .... (b) The ``ELBO'' in the tables should be negative. (c) Some typos in the theorems proof: e.g., proof of theorem 3.2, at the very bottom of page 12, "Replace $b - \gamma$ by $V$'', $b$ and $\gamma$ should be switched over....
>
> **Response:** (a) and (d) we will fix all the typos in the revision. (b) Thanks for pointing this out. We are actually calculating the negative log likelihood (NLL). (c) We should indeed switch $b$ and $\gamma$ in the proof.
>
>
>
> **Reviewer question 1:** classifier-based guidance v.s. time-varying-diffuser.
>
> **Response:** In the classifier-based guidance, we apply the gradient to drive the trajectory points outside the unsafe set whenever they enter the unsafe sets. As the trajectory points frequently enter the unsafe sets due to the lack of set invariance property (while the CBF method does have), applying classifier-based guidance at all times could mess up the diffusion. As a result, the  classifier-based guidance method would fail to identify the constraints and thus fail to satisfy them as there is no explicit constraint information involved (the CBF method does have since it evaluates the derivative of the safety constraints along the diffusion dynamics).
>
> **Reviewer question 2:** I have a hard time to understand what definition 2.3 Control Barrier Function...
>
> **Response:** The CBF method actually relies on the Nagumo’s theorem. In other words, the gradient $\dot b$ is non-negative whenever $b = 0$. Then, we can ensure that $b \geq 0$ is always guaranteed if $b(0) \geq 0$.
>
> **Reviewer question 3:** The proof of theorem 3.2
>
> **Response:** This is because in the last inequality(the second inequality) $\dot x_k^j$ is the derivative of $x_k^j$ with respect to the diffusion time $T$. In other words $\dot x_k^j = \frac{x_j^j}{dT}$. Now we have  that the second and the third inequalities are equivalent. The diffusion time T is the time defined in the diffusion dynamics in (6) or (7). It might be better to use other notation instead of $T$ to avoid the common sense that $T$ is a terminal time.
>
> The robust term is to ensure the strict satisfaction of the safety constraints due to the noise and asymptotic property of the CBF, as shown in the proof of theorem 3.2.
>
> **Reviewer question 4:** Experiments details for robust function $\gamma$, $\alpha$ and $r$ in equation 14?
>
> **Response:** In the experiments, the robust function $\gamma$ is set to 0.01 according to theorem 3.2 and the given $N$ and $\varepsilon$. The extended class function $\alpha$ is a linear function with coefficient 1. The $r$ in equation 14 is a relaxation (slack) variable that makes the CBF constraint (equation 9) soft, and thus make the optimization always feasible. It is to be determined in the optimization (equation 14). This is a common trick to make an optimal control problem always feasible.
>
> **Reviewer Ethics Concerns:** The subsections Diffusion Probabilistic Models and Notations in sections 2 Preliminaries have significant similarities to subsection 2.2 in section 2 Background from the paper ....
>
> **Response:** We rewrote this subsection to address this problem. Please see the revised manuscript.